# Evaluation of bacterial proliferation with a microfluidic-based device: Antibiochip

**Valentina Gallo[1], Alessia Ruiba[2], Massimo Zanin[1], Paolo Begnamino[1], Sabina Ledda[3], Tiziana Pesce[3], Giovanni Melioli[2,3,4], Marco Pizzi** [1] *

**1** Biomed & Nanotech Development & Business Unit, ELTEK S.p.A., Casale Monferrato (AL), Italy,
**2** Phenomix S.r.l., Genova, Italy, **3** Microbiology Section, Laboratorio Albaro S.r.l., Genova, Italy,
**4** Department of Biomedical Sciences, Humanitas University, Pieve Emanuele (MI), Italy

* m.pizzi@eltekgroup.it

**Data Availability Statement:** All relevant data are within the manuscript and its Supporting Information files

**Funding:** MP, VG, MZ and PB are employees of ELTEK S.p.A and developed the microchannels and

## Abstract

The measurement of the proliferation (and the relevant inhibition of proliferation) of microbes is used in different settings, from industry to laboratory medicine. Thus, in this study, the capacity of the Antibiochip (ELTEK spa), a microfluidic-based device, to measure the amount of *E. coli* in certain culture conditions, was evaluated. An Antibiochip is composed of V-shaped microchannels, and the amount of microparticles (such as microbes) is measured by the surface of the pellet after centrifugation. In the present study, different geometries, volumes and times were analyzed. When the best conditions were identified, serial dilutions of microbial cultures were tested to validate the linearity of the results. Then, with the use of wild *E. coli* strains isolated from medical samples, the relationship between bacterial susceptibility to antibiotics measured by standard methods and that measured by the Antibiochip was evaluated. In this report, the good quality performances of the methods, their linearity and the capacity to identify susceptible microbial strains after 60 minutes of incubation are shown. These results represent a novel approach for ultrarapid antibiograms in clinics.

## Introduction

The evolution of medical diagnostics implies that more sensitive, reproducible and rapid methods are developed. Indeed, severe medical pathologies (e.g. represented by cancer, transplant, poly-traumatized patients and immature newborns) require a real time therapeutic intervention for a reliable management of the patients. Notably, even less severe patients must be diagnosed and treated very rapidly in order to reduce the absence from work or school and the risk of complications. All these features result in a worsening of the health and the quality of life of patients and in an increased cost for the patients and/or for the community.

Classic microbiology tools have been developed in the last decades. These include the identification methods and the tests for the evaluation of the susceptibility to antibiotics. Notably, at least for the European countries, the latter has been strictly coded by the rules of Eucast, coding which antibiotic should be tested for each microbe in different clinical situations. Practically, classic methods use a biochemical classification for identification of microbes and *in vitro* proliferation test for Antibiotic Susceptibility Testing (AST).

the centrifuge in their plant, with the support of ELTEK funding. GM is a co-founder and scientific director of Phenomix ltd, AR is a biotechnologist working as research assistant in Phenomix ltd. Phenomix was partially granted by ELTEK as research contractor and partially supported the study with reagents, materials and human resources. SL and TP are employees of a private laboratory, belonging to a multinational organization that allowed the use of its microbiology unit for studies of the prototypes with human pathogens. No other funding or support was available. We can confirm that "The funder provided support in the form of salaries for authors [MP, VG, MZ and PB], but did not have any additional role in the study design, data collection and analysis, decision to publish, or preparation of the manuscript. The specific roles of these authors are articulated in the 'author contributions' section".

**Competing interests:** MP, VG, MZ and PB are employees of ELTEK S.p.A and developed the microchannels and the centrifuge in their plant, with the support of ELTEK funding. GM is a co-founder and scientific director of Phenomix ltd, AR is a biotechnologist working as research assistant in Phenomix ltd. Phenomix was partially granted by ELTEK as research contractor and partially supported the study with reagents, materials and human resources. We confirm that our affiliation does not alter our adherence to PLOS ONE policies on sharing data and materials.

Classic microbiology provides an overnight (18–24 hours) incubation before identifying and characterzing the microbes. Thus, virtually all bacterial strains characterized by a (residual) proliferative activity can be detected by using visual inspection (turbidity) or more accurate instrumental measurements such as turbidimetry and rephlectometry only. Notably, microbiologists had only rarely focused their attention to the early phases of bacterial proliferation. Indeed, microbiology manuals describe the first 6 hours as a steady state phase where no proliferation can be observed.

However, an overnight culture time, despite very convenient for an easy detection of the proliferation, seems to be too long for patients in severe medical conditions. For this reason, other more rapid methods have been developed and have entered in the clinical microbiology routine. Nevertheless, the problem of increasing the speed of microbiological diagnosis still exists and rumors are circulating that many researchers and industries are very active on it. At present, the new techniques based on realtime PCR, MALDI-TOF [1] have reduced the Turn Around Time (TAT) to hours. In very recent period, other even more rapid methods based on infra-red spectrometry have been developed for bacterial ID [2, 3], reducing the TAT to minutes. However, for AST, the new and sometimes complex and expensive methods have only reduced the time to few hours. Thus, a significant reduction of the TAT of the AST procedures is mandatory.

During a series of studies focused on the possible uses of micro- and nanochannels in laboratory diagnostics, researchers from ELTEK S.p.A. (Casale Monferrato, Italy) developed and patented a system characterized by an array of microchannels, in which the quantitation of the particulate fraction of a suspension is measured by optical means. The method is based on different V-shaped microchannel arrays (VSMCAs). When microchannels are filled with a suspension containing a particulate fraction, after one or more rounds of centrifugation, the surface occupied by the pellet is evaluated by a microscope and an image analyzer. Interestingly, in the first industrial tests, this method was sensitive and reproducible that small modifications of the fraction of the particulate could be detected. As a consequence, VSMCAs were tested with microbes. In this report, we describe the first results of bacterial quantitation obtained using wild strains of the human pathogen *Escherichia coli* and some preliminary results of AST performed on an antibiotic panel.

## Materials and methods

### The microchannel array

The VSMCs were built on a plastic disk (diameter 90 mm) by standard photolithographic techniques. Fig 1 shows the different geometries used, with a focus on the different angles of the VSMCs. In the present set of experiments, the microchannels had a length of 20 mm and a height of 11, 22 or 36 μm. The geometry of the microchannels was defined starting from the industrial feasibility of low-cost fabrication techniques and the filling properties of the channels. Thus, 50 μm wide channels were identified as a suitable starting point. Fig 2 shows the loading sector of the microchannels (A), the array of microchannels (B), the microbes floating into the microchannels before centrifugation (C) and the microbial pellet after centrifugation (D). To evaluate the quality of the pellets obtained by these different angles and volumes, the following parameters were kept under consideration: 1) the filling time of the channels, important to increase the speed of the test time; 2) the shape of the microbial pellet after centrifugation, important to maintain a regular geometric shape (triangle-like shape) that can be measured by the image-analysis software; and 3) the texture of the microbial pellet after centrifugation, important for the stability of the pellet during the time between centrifugation and the measurement.

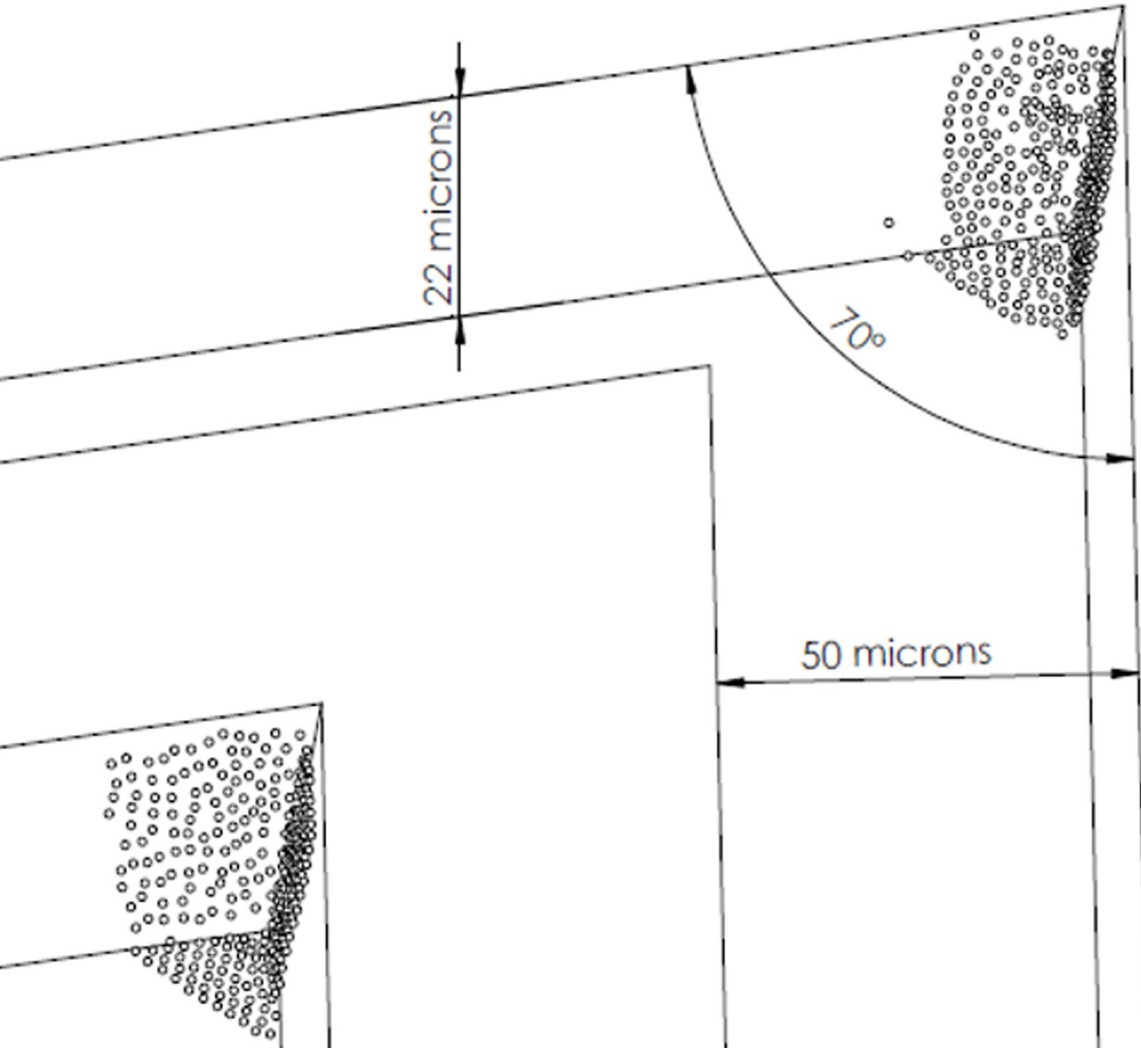

**Fig 1. Examples of different geometries and shapes designed and tested during the industrial phase of the development of the VSMCs.**

### Microorganisms

The microorganisms used in this study were *E. coli* isolated from clinical samples (Table 1). In detail, the bacterial strains used in this study were obtained from clinical microbiology laboratories. receiving biological samples from hospital and community infections. Colonies from isolation plates were further isolated, grown overnight in broth, and then a master microbe bank (MBB) was established and stored at -20˚ C in 20% glycerol in saline solution. The bacterial concentration was measured by turbidimetry in McFarland units. In the course of a preliminary set of experiments, McFarland units were compared with CFU/mL and, in our experimental conditions, 1 McFarland corresponded to $1\times10^9$ E. coli. From that moment, all cultures were measured in turbidimetry. Further subcultures of the bacterial strains were conducted starting from one vial of the MBB. Of all bacterial strains, the identification of the different strains as well as the antibiotic susceptibilities were obtained by a Biomerieux Vitek® 2. Some strains were susceptible to all antibiotics tested, while others had resistances (Table 1).

The list of antibiotics used in the esperiments are reported in Table 2.

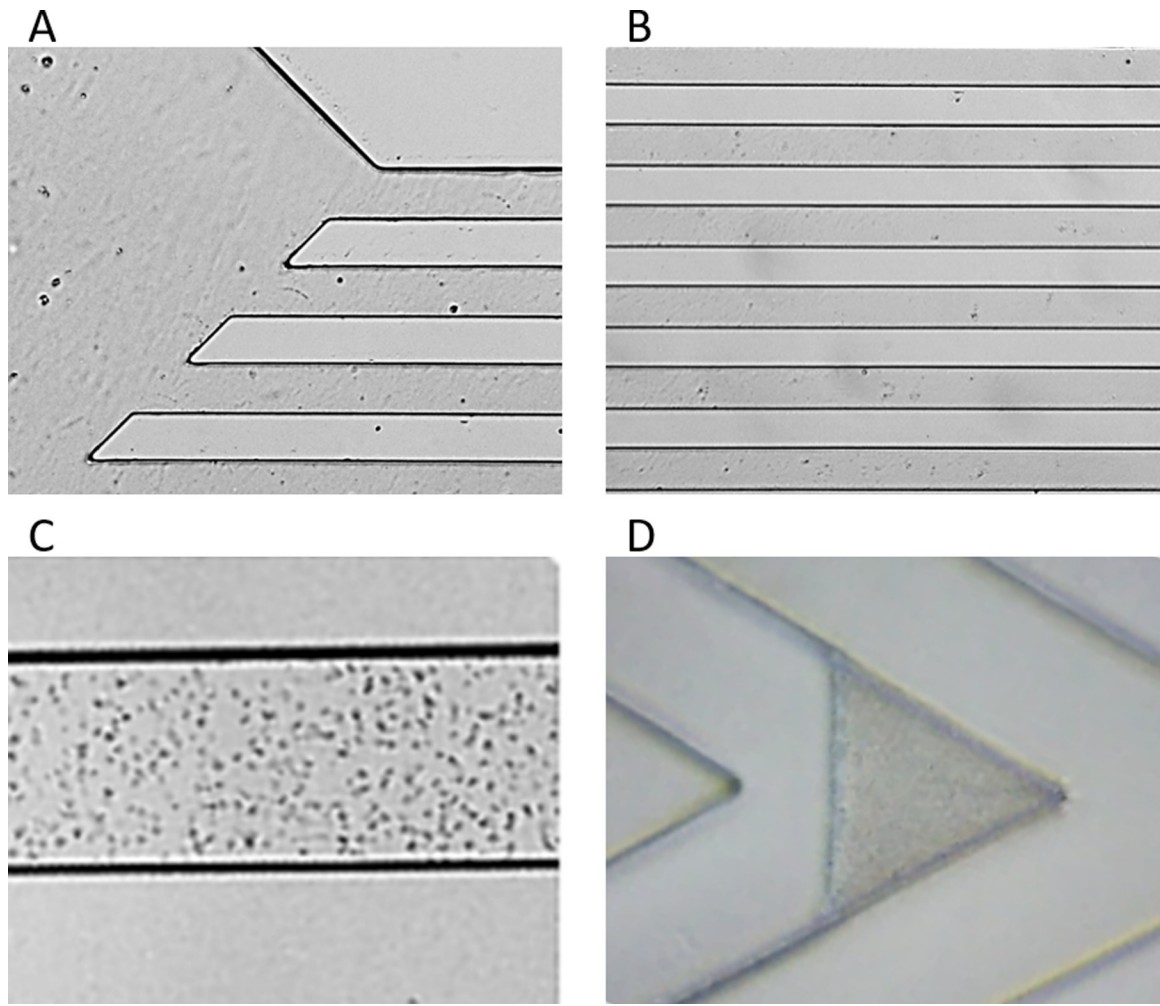

**Fig 2. VSMCs at work.** (A) Loading sector of microchannels, (B) the array of microchannels, (C) microbes (*E. coli*) floating in the microchannels before centrifugation and (D) the microbial pellet after centrifugation.

**Table 1. Microorganisms used in the study.**

| # | ID | GEN | | CIP | | CRO | | SXT | | AMC | | FOF | | FEP | | AMP | |
|---|---|---|---|---|---|---|---|---|---|---|---|---|---|---|---|---|---|
| | | MIC | Int | MIC | Int | MIC | Int | MIC | Int | MIC | Int | MIC | Int | MIC | Int | MIC | int |
| 1 | SL01Eco | < = 1 | S | <0.25 | S | < = 1 | S | < = 1 | S | < = 2 | S | <0.16 | S | <1 | S | 32 | R |
| 2 | SL08Eco | < = 1 | S | <0.25 | S | < = 1 | S | < = 1 | S | < = 2 | R | <0.16 | S | <1 | S | <2 | S |
| 3 | SL39Eco | < = 1 | S | <0.25 | S | < = 1 | S | 8 | R | < = 2 | S | <0.16 | S | <1 | S | <2 | S |
| 4 | SL54Eco | < = 1 | S | <0.25 | S | < = 1 | S | < = 1 | S | < = 2 | S | <0.16 | S | <1 | S | <2 | S |
| 5 | SO17Eco | 8 | R | <0.25 | S | 8 | R | 8 | R | >32 | R | <0.16 | S | <1 | S | <2 | S |
| 6 | SO18Eco | 16 | R | <0.25 | S | 8 | R | 16 | R | >32 | R | <0.16 | S | >64 | R | <2 | S |
| Breakpoint (µg/mL) | | 4 | | 1 | | 2 | | 4 | | 8 | | 32 | | 4 | | 8 | |

GEN: Gentamycin; CIP: Ciprofloxacin; CRO: Ceftriaxone; STX: Trimethoprim-sulfamethossazol; AMC: Amoxicillin-clavulanic acid; FOF: Fosfomycin; FEP: Cefepime.

All antibiotics were used at the breakpoint according to Eucast (http://www.eucast.org/clinical_breakpoints/).

## Measurement of microbial concentration by VSMCs

For all experiments, in order to mimic the real life situation, a vial of each microbe, stored at -20˚C in glycerol as already stated, was thawed and "classically" plated by a sterile loop in a CLB (E.coli selective) agar plate (Bio-Merieux). This step is highly similar to the first isolation of a microbe from a clinical sample. Microbes were harvested by collecting one or more colonies with an inoculating loop from the agar plates where they were isolated. The colonies were dissolved in 0.5% broth (Oxoid BC0102M), and the bacterial concentration was measured in McFarland units with a Bio-Merieux densitometer. The bacterial suspension was used when the McFarland number was 1 or more.

In a first series of tests, 0.005 to 0.01 mL of two-fold dilutions of each microbe were loaded into the microchannel pre-chamber. When the microchannels were filled with the suspension, disks containing the VSMCs were centrifuged at 7000 rpm, corresponding to 1972 g, for 3 minutes, according to the operating conditions identified in the preclinical (industrial feasibility) phase of the study. Of note, during the preclinical/industrial phase of development, only yeasts were used for safety reasons. On the contrary, in the second (clinical-like) phase of the study, tests were performed in two microbiology authorized microbiology laboratories. For this set of experiments, different E coli strains were plated in VSMCs built with different angles and volumes.

To detect the presence of the microbial pellet and to calculate the surface of the pellets, three prototypes were developed in ELTEK and, after a technical validation, one remained in the industry for further tests and development, while the others were placed in two clinical and/or experimental microbiology laboratories. The prototype was basically a high speed centrifuge that could insert the VSMC containing disk on its axis. An electronic rheostat and a RPM counter allowed the control of the speed. For the measurement of the microbial pellet, a USB-driven 800X magnification objective (AM7515MT8A Dino-Lite Edge Microscope, from AnMo Electronics Corporation, Hsinchu, Taiwan) was assembled in the centrifuge to measure the bacterial pellets. A 3D electric servomechanism (ELTEK Spa) was used to align and focus the lens on the VSMC. A panel of pictures of the surface of the microchannel occupied by the microbial pellet was collected and stored into a laptop computer. The surface of the pellets were measured on the different pictures by ImageJ, a free NIH software for image analysis. Finally, data collected on pellet surfaces were stored in a mySQL dedicated DB stored in a "cloud" server. These date were further analyzed by using different statistical methods, using MS-EXCEL and PAST, a powerful freeware statistics analysis tool [4].

Using the same procedure, another series of tests was carried out to evaluate whether VSMCs were suitable to detect not only the differences in microorganism concentrations obtained by a serial dilution but also the inhibition of bacterial proliferation caused by antibiotics in susceptible strains. For this, different microbe strains (Table 1) were incubated in a plastic tube at 37˚C in a moist chamber in the presence of the breakpoint concentration of different antibiotics. After 30 and 60 minutes, 0.005 to 0.01 mL of the culture was collected by a micropipette and loaded into the loading chamber of the VSMC. When the VSMCs were filled (in general, after less than 3 minutes, the volume dispensed at the entrance of the V shaped microchannels was in any case in excess respect to the microchannels volume), the disks were centrifuged, and the pellets were detected and measured as described above. In this series of tests, the above described *E. coli* strains, with different susceptibility to antibiotics were used.

**Table 2. Antibiotics.** The following antibiotics were used.

| Antibiotic | Source | Form | Original | Preparation | Breakpoint |
|---|---|---|---|---|---|
| Amoxicillin/acido clavulanico | DOC Generici | Powder | 875 mg + 125 mg | Solved in H2O | 8 μg/mL. |
| Trimethoprim-sulfamethossazol | Roche | Tablets | 160 mg + 800 mg | Minced and solved in H2O | 4 μg/mL |
| Gentamicin | MSD | Vials | 40 mg/mL | Diluted in H2O | 4 μg/mL |
| Fosfomicin | Doc Generici | Powder | 3 g | Solved in H2O | 32 μg/mL |
| Ciprofloxacin | DOC Generici | Tablets | 250 mg | Minced and solved in H2O | 1 μg/mL |
| Ceftriaxone | Roche | Vial | 125mg/mL | Diluted in H2O | 2 μg/mL |
| Cefepime | Bristol-Myers Squibb | Vial | 330 mg/mL | Diluted in H2O | 4 μg/mL |
| Ampicillin | Biopharma | vial | 250 mg/mL | Diluted in H2O | 8 μg/mL |

## Results

Different geometries and volumes were used during the validation phase of the VSMC to define the best configuration of the microchannels. Table 3 shows the results of these tests. The thinnest channels (11-μm-thick) required a long filling time and did not guarantee a proper concentration in the channels when compared with the 22- and 36-μm-thick channels. Similarly, 70˚ angles were the best, while 90˚ pellets were unstable and 60˚ pellets displayed several bubbles in the channels.

Due to a different diffusion behavior, even if the size of the channels was approximately 10 times the size of the particles, the concentration inside the thinnest channels did not reproduce the concentration in the sample, as shown in Fig 3A. Comparing the absolute values of the pellet areas, the behavior of the 22- and 36-μm-thick channels was very similar, while the 11-μm-thick channels showed a lower particle concentration. This is a well-known phenomenon due to anomalous diffusion when the channel size and particle size are not very different [5, 6].

The concentration measurement with 22- and 36-μm-thick channels (Fig 3B and 3C) was comparable, but the pellet in the 36-μm-thick channels was unstable, leading to a larger area distribution at higher concentrations.

The correlation between the expected and measured relative concentrations of the samples was correct for the 22- and 36-μm-thick channels, while it was not respected for the 11-μm-thick channels, as shown in Fig 4.

Based on these results, subsequent experiments were conducted with the 22-μm-thick VSMC at a 70˚ angle.

## Dilutions

Several tests were performed using two-fold serial dilutions of a bacterial suspension starting from an optical density > 1 McFarland. Fig 5 shows the results of these assays. It is evident

**Table 3. Evaluation of geometries and volumes to define the best configuration of microchannels.**

| Parameters evaluated | Thickness of the microchannel (in μm) | | | Angle of the V-shaped channels (in degrees) | | |
|---|---|---|---|---|---|---|
| | 11 | 22 | 36 | 90˚ | 70˚ | 60˚ |
| Filling of the channels | - | + | ++ | + | + | - (2) |
| The shape of the pellet after centrifugation | +/- | ++ | + (1) | + (1) | ++ | - (2) |
| The texture of the pellet after centrifugation | +/- | + | +/- | + (1) | ++ | - (2) |

(1) Instability of the pellet.

(2) Presence of air bubbles in the microchannel.

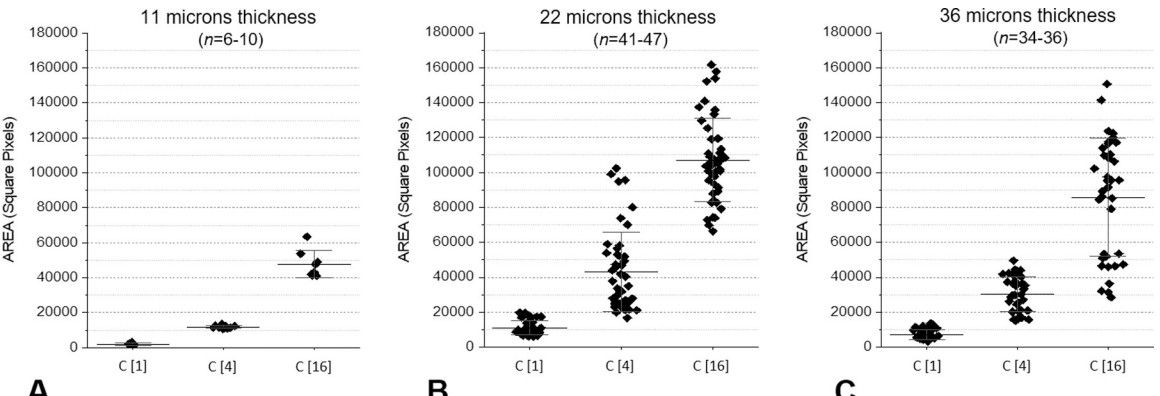

**Fig 3.** Diffusion of microbes in VSMCs with different thicknesses: (3A) 11μm, (3B) 12μm, (3C) 36μm. Measurement of the areas in three different microbial concentrations (1:4 dilutions). See also the text.

that the surface of the pellets was in strict correlation with the dilutions of microorganisms. In practical terms, these dilutions corresponded to four replications of the original bacterial suspension; considering that *E. coli* doubles in 20–30 minutes, a 16-fold increase is representative of a culture time from 1.2 to 2 hours. To further explore the capacity of VSMCs to detect a small difference in the microorganism concentration, another series of experiments was carried out using a narrower dilution range, i.e., 1, 1.5, 2, 3 and 4. Fig 6 shows the results of these tests. In these conditions, it was evident that the surface of the pellets in the VSMCs was highly representative of the small differences of the original microorganism suspensions.

## Inhibition of bacterial proliferation by antibiotics

Starting from the abovementioned results and the clear capacity of VSMCs to detect small differences in the concentration of microorganisms in a suspension, several tests were planned to verify whether the inhibition of proliferation due antibiotics could be measured when different antibiotics were introduced to susceptible microorganisms. For this, different wild strains of *E. coli*, obtained from clinical samples and fully characterized by standard microbiology methods, were incubated with different antibiotics (listed in Table 2) at the respective breakpoint values. Fig 7 shows an example of the experiments performed using bacterial strains (SL1Eco, SL8Eco,

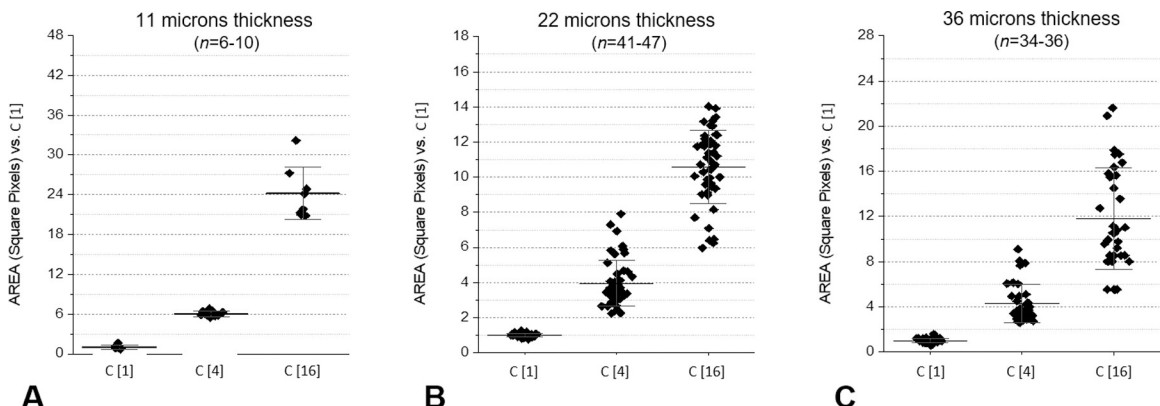

**Fig 4.** Diffusion of microbes in VSMCs with different thicknesses: (4A) 11μm, (4B) 12μm, (4C) 36μm. Correlation of the areas vs expected results in three different microbial concentrations (1:4 dilutions). See also the text.

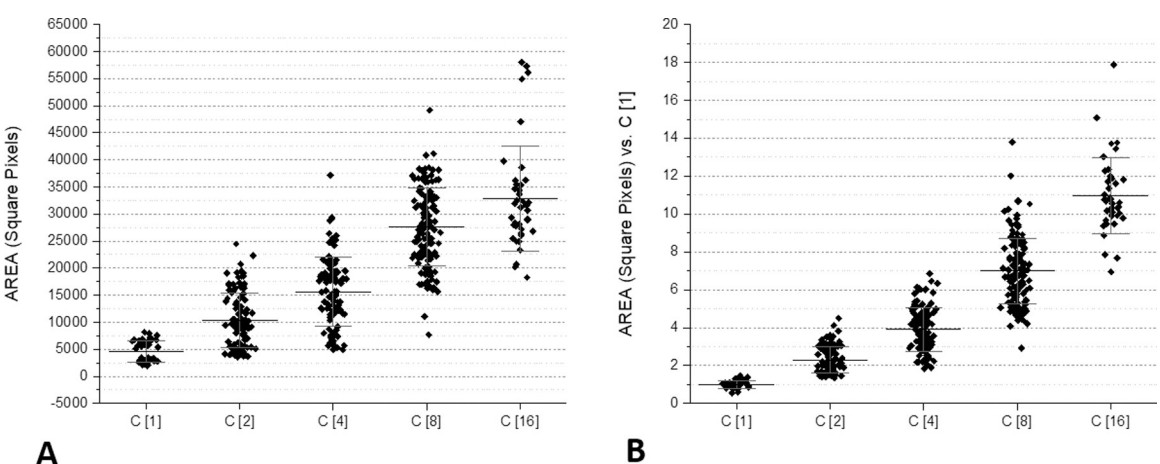

**Fig 5. Microbe areas in VSMCs observed with two-fold serial dilutions of a bacterial suspension.**

SL38Eco) that were susceptible to antibiotics in routine antibiograms. Once clear evidence of the capacity of VSMCs to detect the inhibition of the proliferation of *E. coli* was observed within 60 minutes of culture, other wild-type bacterial strains (SO17Eco and SO18Eco), characterized by resistance to antibiotics (namely, GEN and CRO), were tested and, as expected, different results were observed. Fig 8 shows a representative result.

In an attempt to find some general rule, the results of the VSMCs for different antibiotics were grouped according to the antibiotic resistance as detected by the standard method. A simple statistic (mean and standard deviation after transforming the percentages to arcsin) was calculated. The mean result for strains evaluated as susceptible by standard methods (independently from the antibiotic used) was represented by a capacity of proliferating of 25% (corresponding to a 75% inhibition), with an upper limit of 60% (40% inhibition). For resistant strains (at the MIC), the proliferating capacity was 85% (15% inhibition) with a lower limit of 60%. According to these results, the presence of a proliferative capacity < 60% was considered to correspond with the susceptibility to the standard tests, while the maintenance of a proliferative capacity > 60% of the controls was considered suggestive of antibiotic resistance.

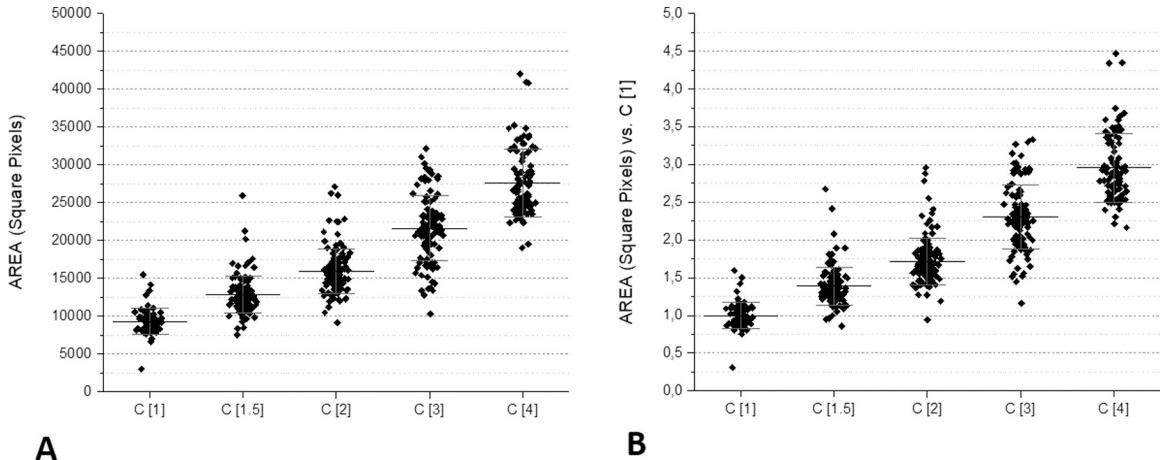

**Fig 6. Microbe areas in VSMCs loaded with a narrow dilution range (1, 0.75, 0.50, 0.25, 0.1).** In these experiments, different McFarland concentrations (from 1 to 2) were used and normalized for dilutions.

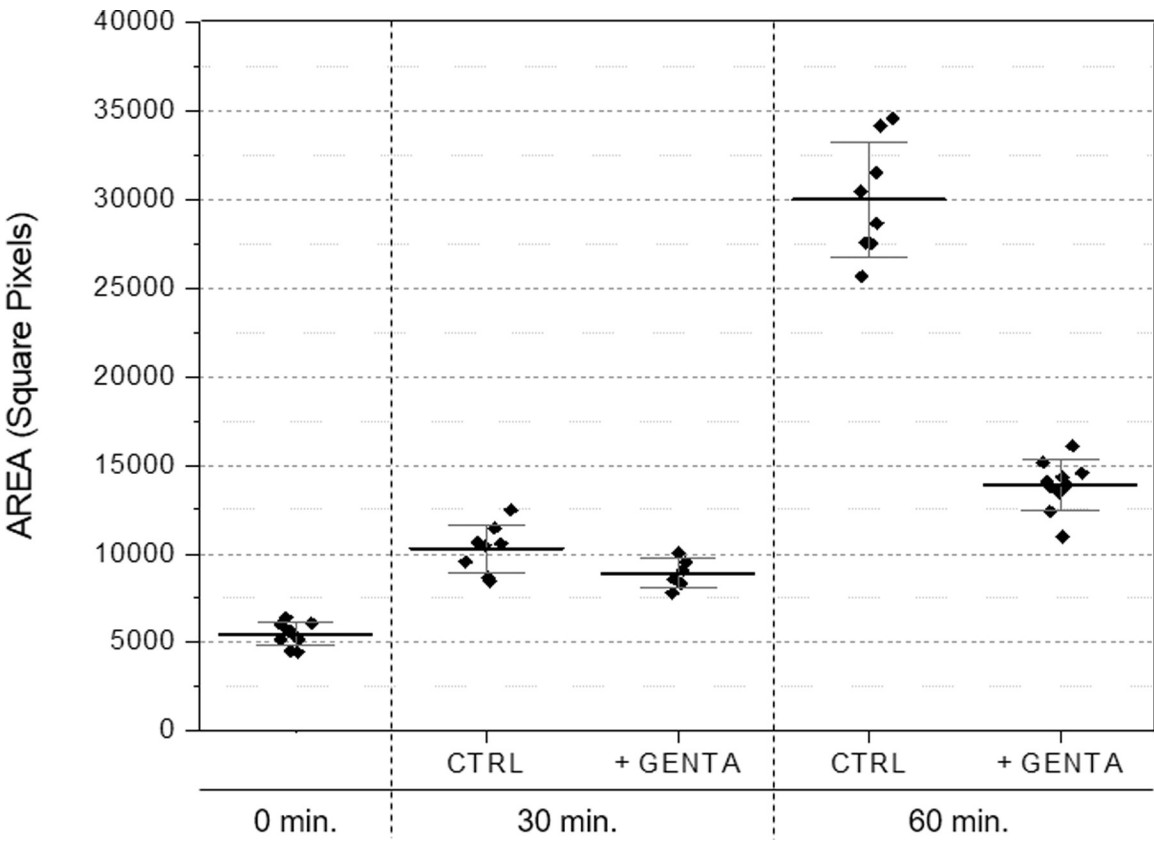

**Fig 7. Inhibition of bacterial proliferation in different strains of *E. coli* susceptible to antibiotics.** The results were obtained after 30 and 60 minutes of incubation. Susceptible strains were barely evident after 30 minutes but clearly evident after 60 minutes.

In another series of experiments, a different selection of antibiotics (namely, GEN, AMP, FOS, CIP, FEP, CTX, CRO at the breakpoint concentrations) were used. A representative result of the strain SL01Eco, resistant to Ampicillin, is shown in Fig 9.

The percentages of growth were trasformed in arcsin and then the distribution of the results was plotted. The mean arcsin value for antibiotics that 0.43 arcsin and the standard deviation was 0.18 arcsin. Thus, adding 2SD to the mean value, a raw cutoff can be calculated, corresponding to 0.79 arcsin. Growth in the presence of AMP was 1.19, largely higher than the cut/off value.

It was however evident that rules of end-point measurement based on proliferation could not be immediely applied to the proliferation of the first hour. For this reason, two GEN sensible and two GEN resistant E. coli strain were repeatedly tested in the presence of decreasing concentration of GEN (4, 2, 1, 0.5 and 0.25 BPs). A representative result is shown in Fig 10, where It is evident that, at least in these experimental conditions, a clear difference of susceptibility can be observed at 4xBreakpoint.

While the large majority of experiments were carried out using *E. coli* (mainly wild-type), other experiments were conducted with other gram-negative bacteria, such as *Enterobacter aerogenes*, *Klebsiella pneumoniae* and *Proteus mirabilis* (not shown). At present, the number of these tests is too small to attempt any statistical consideration, but from a descriptive point of view, these other Bacteriaceae also had a behavior comparable to that of *E. coli* in the VSMC model. In the same series of experiments, other incubation times were also considered, in particular 30 minutes and 120 minutes. The 120-minute incubation time confirmed the results

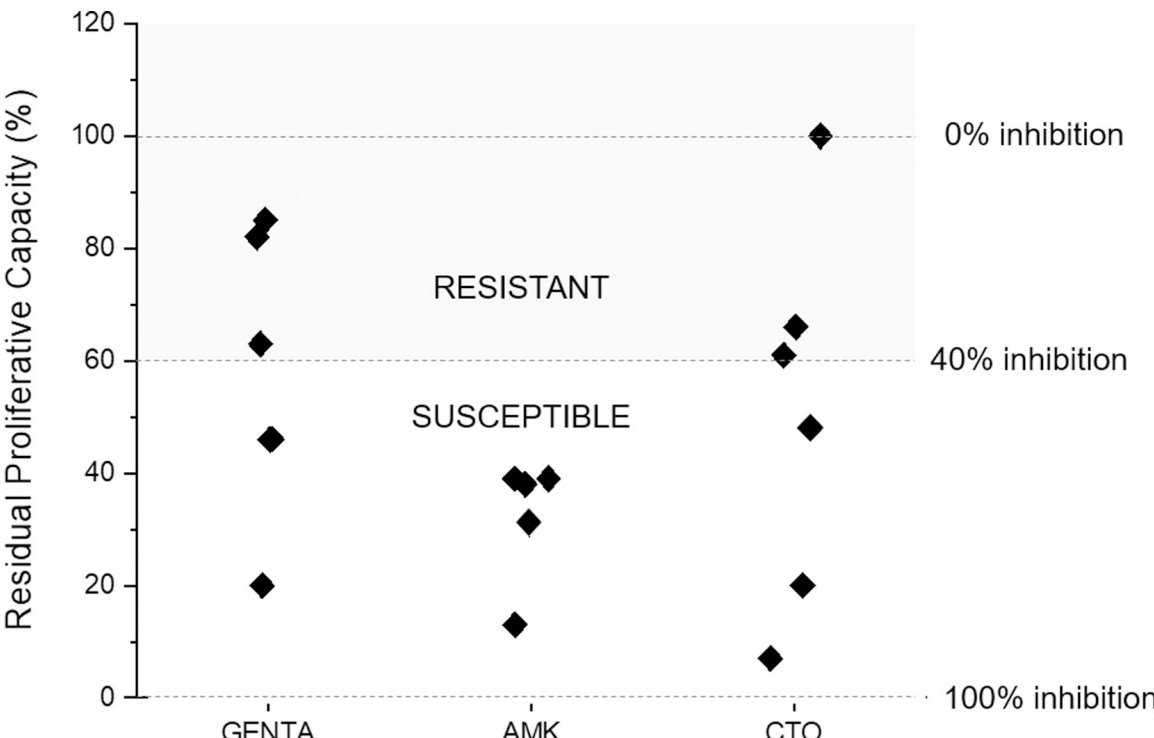

**Fig 8. Analysis of *E. coli* strains cultured in the presence of antibiotics; 3/5 were resistant to GEN, all were susceptible to AMK and 3/6 were resistant to CTO in standard microbiology tests.** VSMC results correlate well with standard results.

observed at 60 minutes, and for this reason, at least in the present setting of assays, no further tests were carried out. In contrast, highly promising results were observed after 30 minutes of culture in the presence of antibiotics; nevertheless, the number of tests performed does not seem to be sufficient to provide some shareable conclusions. Finally, to evaluate whether both the instrument prototypes and the whole method were suitable for a practical use, some experiments, including non pathogens, were carried out in the ELTEK plant, while other tests, providing the human pathogen E coli, were performed in a clinical microbiology unit and in an experimental microbiology unit, in two different town. Results were high superimposable (not shown).

## Discussion

Methods to measure the capacity of an antibiotic to inhibit the proliferation of a bacterial strain have been developed in the past, starting from the first description by Fleming [7] to the antibiotic diffusion in paper disks [8] and the more recent measurement of the minimal inhibitory concentration [9]. The recently introduced technology by Accelerate Diagnostics uses morphokinetic cellular analysis to measure the capacity of a microbe to proliferate in the presence of an antibiotic [10]. At present, there is evidence that increasing the speed of the detection of antibiotic resistance is a real topic [11], [12, 13], [14], [15].

In routine clinical microbiology settings, the time to measure antibiograms from an automated or manual system ranges from 6 to 12 hours. In some cases, for example, with the use of Accelerate or by the use of the Alfred 60AST (Alifax, Nimis, IT)[16] culture system, results can be observed in a shorter time. Of note, this "laboratory time" is added to the period (usually, one night) used to isolate a microbe from a biological sample. The rapid methods currently

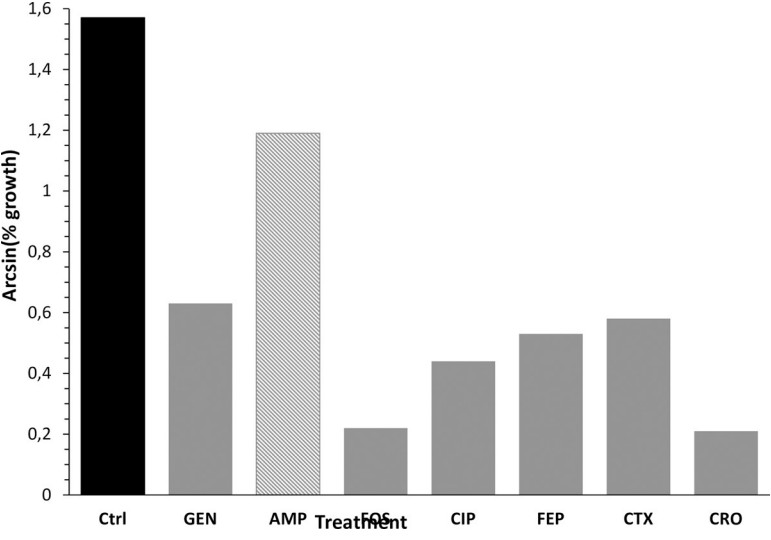

**Fig 9. Results of the strain SL01Eco, resistant to Ampicillin.**

available also require a specific laboratory organization possible only in highly specialized structures. Nevertheless, patients with infectious diseases or patients with severe systemic diseases complicated with a concomitant infection require a very rapid evaluation of the susceptibility of the microbe to start a specific antibiotic therapy in a very short time. This medical emergency is complicated by the diffusion of antibiotic resistance [17] and by the fact that new antibiotics do not seem to be in the industrial pipelines [18].

The VSMC method described above seems to be different from others and highly innovative. Indeed, the tests described in this work show that the surface of the pellet, strictly related to the number of microbes, is detected rapidly and accurately. The reproducibility is good, and the correlation between measured surfaces and expected results–at least based on two-fold dilution assays–is excellent. In addition, other more accurate measurements (performed using a 25% increase in the bacterial concentrations) also showed that the capacity to distinguish between apparently small differences is good. This seems to be even more relevant considering that the prototypic gram-negative microbe *E. coli* doubles in 20–30 minutes. Thus, the increase in the number of microbes–paralleled by the increase in the surface of the pellets–ranges between 2- to 3-fold in a period of one hour. Notably, an increase in the number of microbes measured using VSMC has also been detected in 30 minutes in some preliminary but reliable experiments. These results can be observed by starting the microbial culture with a concertation of microbes ranging from 0.5 to 1 McFarland, a routine concentration of microbes for automated clinical tests, easily achievable by harvesting more than one colony from the isolation plate. Therefore, the method based on VSMC, herein described, seems to be able to identify antibiotic-susceptible microbes in laboratory tests lasting a period of time (namely, 60 minutes) significantly shorter than other routine tests used in clinical microbiology.

The two (classic and VSMC-based) approaches have been compared in different experimental conditions. First, few antibiotics were used to distinguish between susceptible and resistant strains. Then, a wider selection of antibiotics, resembling the panel provided by EUCAST for E. coli, was also tested. Even if an extremely simple statistical approach was used, a clear relashonship between antibiotic susceptibility obtained by the two methods, was observed. However, as soon as many other results are obtained using different microbes and

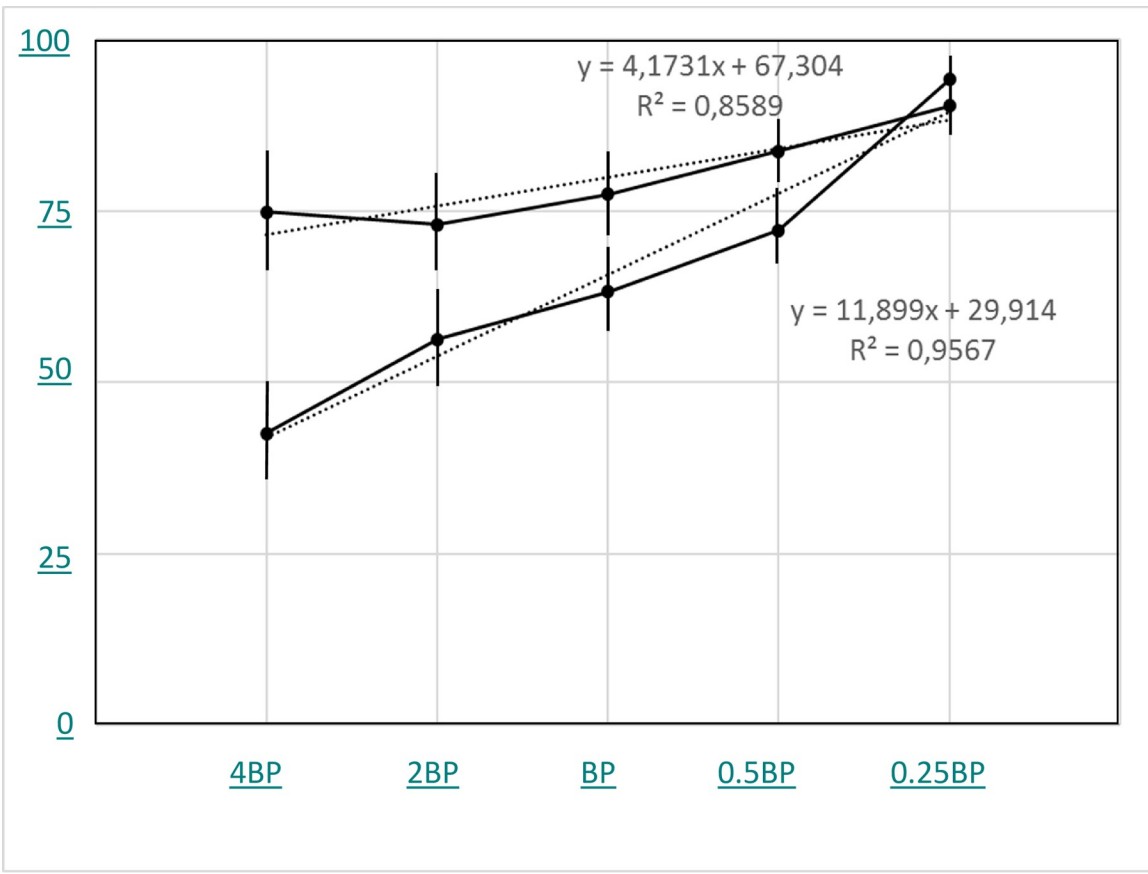

**Fig 10. Comparison of GEN sensible and GEN resistant *E. coli* strains, tested in the presence of decreasing concentration of GEN (4, 2, 1, 0.5 and 0.25 BPs).**

different antibiotics, a much more sophisticated statistical analysis will be performed, but, at present, data seem to be robust. The antibiotic doses suitable to distinguish susceptible from resistant strains have a central role in the development of the VSMC-based AST. Indeed, a good correlation was observed in assays where the BP concentration of each antibiotic was used. But in the specific experimental conditions, the difference between susceptible and resistant strains is the result of a statistical analysis. On the contrary, in classic endpoint assays lasting few hours to overnight, resistant strains are fully grown and susceptible strain cultures are sterile. In the attempt to identify the antibiotic doses to be used in the VSMC, serial dilutions of antibiotics were carried only for GEN on 2 susceptible and 2 resistant E coli, VSMCs seem to be suitable in measuring the MIC of single antibiotics. That a clear distinction of resistant from susceptible strain can be achieved using 4BP is the result of a preliminary series of experiments. Probably, in different conditions with different antibiotics, the concentration of Antibiotic could be different. However, it seems to be relevant that the BP used in in vitro end-point tests does not seem to be optimal for the 60 minutes test: indeed, susceptible stains still proliferate and resistant strains are partially inhibited. In this context, the trough minimal levels for GEN are 0.5–2 microgr/mL and the peak GEN levels are 5–10 micrograms/ml. These concentrations can be maintained in vivo for a relatively short period of time. In other words, MIC seems to be a more "in vitro" than "in vivo" relevant measurement and the results of these experiments with VSMC seems to be more adherent to real life.

A basic medical rule is to administer patients with an antibiotic that may have a high probability of being efficient in controlling the infection. Resistances are extremely relevant from an epidemiological point of view, but in the real time management of the patient, they seem less relevant.

In the present experimental phase of analysis of microbial proliferation (and inhibition) in VSMCs, specific warnings must be given. First, for microbes that were resistant to standard clinical methods, a proliferation identical to that observed for the controls was never observed. In other words, even for "resistant strains", a certain susceptibility is detected in short time tests. Second, antibiotic susceptibility can be identified with great accuracy, even if a certain residual proliferative capacity is detected in 60 minutes of incubation. In these conditions, the VSMC-based method detects something that is invisible in the long-term culture (>6 hours) used by the standard methods, where antibiotics have plenty of time to kill all microbes.

Third, antibiotic resistance seems more difficult to define in very short-period assays. Indeed, a certain inhibition is always detected, even if a correlation between certain behavior patterns of "resistant" microbes measured by VSMC and "classic" routine tests was observed. For example, the introduction of a cut-off of 30% (i.e., resistant microbes in the presence of antibiotics maintained at > 70% capacity of proliferation when compared with the "negative control") indicates that even at an MIC of 1, a certain inhibition of resistant microbes is observed in the short test. This is in partial contrast with the resistant/susceptible paradigm of clinical microbiology. In the same context, one may argue that one of the postulates of clinical microbiology is that a single resistant microbe in a colony of susceptible microbes means that the whole colony should be considered resistant. This can be easily observed in long-term (such as overnight) cultures of microbes, where it is virtually impossible to detect the fraction of resistant microbes in a colony, even if a single mutation is possible but not very frequent. When short time tests are used, the rapidity of the results to the clinician is counterbalanced by the objective difficulty to identify extremely rare microbes carrying a resistance phenotype in the context of a colony. Clearly, this is not a specific problem of the method described in this work, but it is in common with all other attempts to increase the speed of antibiograms. However, even if the problem is considered from a medical point of view, it may be expected that the very large majority of susceptible microbes are kept under control by the antibiotic treatment and, probably, the remaining resistant bacteria can be controlled by an immediate immune-response that has a relevant role in managing bacterial infections. Nevertheless, the presence of a selective agent (the antibiotic) administered to the patient cannot rule out that resistant microbes may proliferate.

The above described ultra-rapid AST tool is even more necessary nowadays: indeed, rapid methods for bacterial identification (such as the well-established MALDI-TOF procedures, the rapid Real time PCR and infra-red spectroscopy) are available, and a significant reduction of AST time is mandatory. Genes associated to antibiotic resistance have been extensively described and laboratory tests for detecting these genetic traits are available. Nevertheless, trivially speaking, not all genes are expressed and, unless some relevant discovery, no data related to the presence of a specific (array of) genes are available to define the dose of antibiotic that inhibits the proliferation of the microbe.

In conclusion, many attempts have been made to reduce the time of incubation of microbes and to detect microbial susceptibility or resistance to antibiotics. In the present study, very short incubations (60 minutes), unimaginable a few years ago, have been shown to be possible to detect microbial susceptibility or resistance to antibiotics, provided that an extremely sensitive method of bacterial proliferation/inhibition is used. However, the clinical impact of this new technology in the control of infectious disease must be carefully evaluated. Obviously, if the described correlation between the VSMC method and classic methods is confirmed,

clinical trials will be much safer to be carried out in complex patients with severe infections to evaluate whether a shorter analytical time has an impact on the patient's clinical evolution.

## Author Contributions

**Conceptualization:** Valentina Gallo, Massimo Zanin, Paolo Begnamino, Giovanni Melioli, Marco Pizzi.

**Data curation:** Valentina Gallo, Alessia Ruiba, Paolo Begnamino, Giovanni Melioli.

**Formal analysis:** Valentina Gallo, Alessia Ruiba, Giovanni Melioli, Marco Pizzi.

**Investigation:** Valentina Gallo, Massimo Zanin, Sabina Ledda, Tiziana Pesce, Giovanni Melioli, Marco Pizzi.

**Methodology:** Valentina Gallo, Sabina Ledda, Tiziana Pesce, Giovanni Melioli, Marco Pizzi.

**Resources:** Marco Pizzi.

**Supervision:** Giovanni Melioli, Marco Pizzi.

**Writing – original draft:** Valentina Gallo, Giovanni Melioli, Marco Pizzi.

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
