## [Decision Letter · Decision Letter 0]

13 Nov 2019

PONE-D-19-26275

Evaluation of bacterial proliferation with a microfluidic-based device: Antibiochip.

PLOS ONE

Dear Dr Marco Pizzi

Thank you for submitting your manuscript to PLOS ONE. After careful consideration, we feel that it has merit but does not fully meet PLOS ONE’s publication criteria as it currently stands. Therefore, we invite you to submit a revised version of the manuscript that addresses the points raised during the review process.

As can be seen by both reveiwers and myself, there is extensive rewriting that is needed. Also the work is very preliminary and more relevant experiments is required. Also to note the methods are not clearly written. It should be that someone else can repeat your method and get similar result. The discussion too has to be relevant to the study and compared with others in the same area.

We would appreciate receiving your revised manuscript by Dec 28 2019 11:59PM. To enhance the reproducibility of your results, we recommend that if applicable you deposit your laboratory protocols in protocols.io, where a protocol can be assigned its own identifier (DOI) such that it can be cited independently in the future. For instructions see: http://journals.plos.org/plosone/s/submission-guidelines#loc-laboratory-protocols

We look forward to receiving your revised manuscript.

Kind regards,

Shamala Devi Sekaran

Academic Editor

PLOS ONE

Journal Requirements:

2. We note that 'The microorganisms used in this study were E. coli isolated from clinical samples'. Please include the reference if the clinical study that the E coli samples were isolated from has already been published.

We note that one or more of the authors are employed by a commercial company: "ELTEK S.p.A." and "Phenomix S.r.l."

Additional Editor Comments:

Please refer to the comments by both reviewers. The writing of the methods and the discussion require extensive rewriting and to be relevant to the study. Methodology needs to be clear.

Reviewers' comments:

Reviewer's Responses to Questions

**Comments to the Author**

1. Is the manuscript technically sound, and do the data support the conclusions?

Reviewer #1: Partly

Reviewer #2: No

2. Has the statistical analysis been performed appropriately and rigorously? 

Reviewer #1: No

Reviewer #2: No

3. Have the authors made all data underlying the findings in their manuscript fully available?

Reviewer #1: Yes

Reviewer #2: No

4. Is the manuscript presented in an intelligible fashion and written in standard English?

Reviewer #1: No

Reviewer #2: Yes

5. Review Comments to the Author

Reviewer #1: Although the technology has clinical impact especially in determining antibiotic susceptibility, the identification of microorganisms still needs the normal culturing Technics. The only advantage is the duration of AST result reporting is shortened, however, this is a very preliminary data and need validation at least with minimum 5 microorganisms with respective standard drugs.

Following are my detailed comments:

Methodology:

In detail, the bacterial strains used in this study were obtained from a clinical microbiology laboratory studying samples from hospital and community infections.- Pls indicate how many strains used.

Measurement of microbial concentration by VSMCs.- The bacterial suspension was used when the McFarland number was 1 or more – This is not clear. Pls indicate in CFU/ml

Pls change centrifugation force units from RPM to g.

In a first series of tests, two-fold dilutions of each microbe were loaded into the microchannel prechamber,and when the microchannels were filled with the suspension, disks containing the VSMCs were centrifuged at 7000 rpm for 3 minutes, according to the operating conditions identified in the preclinical (industrial feasibility) phase of the study. During this set of experiments, different microbes were plated in VSMCs at different angles and volumes.- Pls indicate what microbes are these.

Another series of tests was carried out to evaluate whether VSMCs were suitable to detect not only the differences in microorganism concentrations obtained by a serial dilution but also the inhibition of bacterial proliferation caused by antibiotics in susceptible strains. For this, different microbes were incubated in a plastic tube at 37°C in a moist chamber in the presence of the MIC of different antibiotics- Same here , pls indicate list of microbes with respective concentration (cfu.ml) used.

After 30 and 60 minutes, a microsample of the culture was collected by a micropipette and loaded into the loading chamber of the VSMC. – what is the volume used here?

Results –

No statistical analysis were performed for data presented in Fig 3,4, 5,6,7,8,

Fig 8- implicates that anything below 40% inhibition is resistant, pls explain if this cut-off value is applicable for all bacterial microorganisms. Otherwise indicate that this is applicable only for E coli.

Discussion:

Overall the results implicate that although the technology/method is novel but it is still at very preliminary stage. The identification of bacteria (genus and species) is still not detectable using this method. This should be further elaborated as drawback of this device.

Reviewer #2: 1.Abstract is adequate

2. Introduction - too brief, needs more elaboration

3. Methods: The details of the procedure not included, i.e: starting concentration of the organism, procedure of performing, media used, growth condition, antibiotic susceptibility testing results, etc.

4. Results: How do you compare inhibition rate? The susceptibility of organism is not shown.

The growth curve of the organism is not measured. Therefore, there is no baseline to compare proliferation inhibition rate.

5. Discussion: The discussion is not relevant i.e the comparison with resistant stains (this data was not shown by other methods in the methods and results)

6. PLOS authors have the option to publish the peer review history of their article (what does this mean?). If published, this will include your full peer review and any attached files.

Reviewer #1: No

Reviewer #2: No

---

## [Author Response · Author response to Decision Letter 0]

27 Dec 2019

Dear Dr. Shamala Devi Sekaran,

thank you for considering our paper “Evaluation of bacterial proliferation with a microfluidic-based device: Antibiochip.” worth evaluating. We are submitting a new version of the paper where we tried to address all the points highlighted by the reviewers.

In the following an explanation to specific questions is reported. On the website we will upload also, as required:

• A marked-up copy of the manuscript that highlights changes made to the original version 

• An unmarked version of the revised paper without tracked changes. 

• The additional figures

Best regards

Marco Pizzi

Reply:

Certainly, we will do it

2. We note that 'The microorganisms used in this study were E. coli isolated from clinical samples'. Please include the reference if the clinical study that the E coli samples were isolated from has already been published.

Reply:

Used clinical samples are not coming from clinical studies. Indeed, the very large majority of microbiology tests are performed in out-patients not involved in controlled clinical trials.

We note that one or more of the authors are employed by a commercial company: "ELTEK S.p.A." and "Phenomix S.r.l."

Reply:

MP, VG, MZ and PB are employees of ELTEK S.p.A and developed the micro-channels and the centrifuge in their plant, with the support of ELTEK funding.

GM is a co-founder and scientific director of Phenomix ltd, AR is a biotechnologist working as research assistant in Phenomix ltd. Phenomix was partially granted by ELTEK as research contractor and partially supported the study with reagents, materials and human resources.

SL and TP are employees of a private laboratory, belonging to a multinational organization that allowed the use of its microbiology unit for studies of the prototypes with human pathogens. No other funding or support was available.

We can confirm that “The funder provided support in the form of salaries for authors [MP, VG, MZ and PB], but did not have any additional role in the study design, data collection and analysis, decision to publish, or preparation of the manuscript. The specific roles of these authors are articulated in the ‘author contributions’ section.”

Reply:

We confirm that our affiliation does not alter our adherence to PLOS ONE policies on sharing data and materials. 

MP, VG, MZ and PB as employees of ELTEK S.p.A, founder of the research, performed the research as part of their normal job activity, so no conflict is foreseen. For GM, AR, TP and SL, no competing interests at all as well.

Reply:

Ok, done.

Additional Editor Comments:

Please refer to the comments by both reviewers. The writing of the methods and the discussion require extensive rewriting and to be relevant to the study. Methodology needs to be clear.

Comments to the Author

1. Is the manuscript technically sound, and do the data support the conclusions?

Reviewer #1: Partly

Reviewer #2: No

2. Has the statistical analysis been performed appropriately and rigorously? 

Reviewer #1: No

Reviewer #2: No

3. Have the authors made all data underlying the findings in their manuscript fully available?

Reviewer #1: Yes

Reviewer #2: No

4. Is the manuscript presented in an intelligible fashion and written in standard English?

Reviewer #1: No

Reviewer #2: Yes

5. Review Comments to the Author

Reviewer #1: Although the technology has clinical impact especially in determining antibiotic susceptibility, the identification of microorganisms still needs the normal culturing Technics. The only advantage is the duration of AST result reporting is shortened, however, this is a very preliminary data and need validation at least with minimum 5 microorganisms with respective standard drugs.

Reply:

Being the centrifuge and the micro-channels (the main added values of the method) prototypes non already commercially available, we of course can publish the protocol but in the absence of this tools, researchers could have problems in reproducing the experiments. 

What it should be done could be the installation of the tools in another microbiology lab, where other microbiologists could reproduce the experiments. This is a really unusual procedure in applied research but it can be done. In any case we are working in a further evolution of the system with the objective to increase the user friendliness and accordingly allowing an easier testing in other labs.

Following are my detailed comments:

Methodology:

In detail, the bacterial strains used in this study were obtained from a clinical microbiology laboratory studying samples from hospital and community infections.- Pls indicate how many strains used.

Reply:

OK. A table with the list of microbes was added to the manuscript.

Measurement of microbial concentration by VSMCs.- The bacterial suspension was used when the McFarland number was 1 or more – This is not clear. Pls indicate in CFU/ml

Reply:

We calculated the CFU/mL corresponding to 1 McFarland. In a series of experiments, it resulted to be 0.5 x 109 E. coli/mL

Pls change centrifugation force units from RPM to g.

Reply:

Ok. See the revised version of the manuscript.

In a first series of tests, two-fold dilutions of each microbe were loaded into the microchannel prechamber,and when the microchannels were filled with the suspension, disks containing the VSMCs were centrifuged at 7000 rpm for 3 minutes, according to the operating conditions identified in the preclinical (industrial feasibility) phase of the study. During this set of experiments, different microbes were plated in VSMCs at different angles and volumes.- Pls indicate what microbes are these.

Reply:

OK: In all tests, virtually only Enterobacteriaceae (Gram – bacteria) were used.

Another series of tests was carried out to evaluate whether VSMCs were suitable to detect not only the differences in microorganism concentrations obtained by a serial dilution but also the inhibition of bacterial proliferation caused by antibiotics in susceptible strains. For this, different microbes were incubated in a plastic tube at 37°C in a moist chamber in the presence of the MIC of different antibiotics- Same here , pls indicate list of microbes with respective concentration (cfu.ml) used.

After 30 and 60 minutes, a microsample of the culture was collected by a micropipette and loaded into the loading chamber of the VSMC. – what is the volume used here?

Reply:

See the revised version of the manuscript

Results –

No statistical analysis were performed for data presented in Fig 3,4, 5,6,7,8,

Reply:

A simple statistical analysis is present in these figures, represented by mean and the error bars. However, additional statistical analysis has been added to the manuscript.

Fig 8- implicates that anything below 40% inhibition is resistant, pls explain if this cut-off value is applicable for all bacterial microorganisms. Otherwise indicate that this is applicable only for E coli.

Reply:

OK. See the revised version of the manuscript

Discussion:

Overall the results implicate that although the technology/method is novel but it is still at very preliminary stage. The identification of bacteria (genus and species) is still not detectable using this method. This should be further elaborated as drawback of this device.

Reply:

This is an interesting point raised by the reviewer. However, this set of experiments was carried out in order to really speed the TAT of AST, in particular because for ID, novel and rapid methods (such as Maldi TOF) are now available at diagnostic level.

Reviewer #2: 1.Abstract is adequate

2. Introduction - too brief, needs more elaboration

Reply:

OK. See the revised version of the manuscript

3. Methods: The details of the procedure not included, i.e: starting concentration of the organism, procedure of performing, media used, growth condition, antibiotic susceptibility testing results, etc.

Reply:

OK. See the revised version of the manuscript

4. Results: How do you compare inhibition rate? The susceptibility of organism is not shown.

The growth curve of the organism is not measured. Therefore, there is no baseline to compare proliferation inhibition rate.

Reply:

OK. See the revised version of the manuscript; in particular, the susceptibility of different microbes to ATB using standard tests is shown in Table 2. The inhibition is calculated respect to the growth on the microbe without ATB in 60 minutes.

Growth rate in 1 hour is extremely difficult to be performed, in particular using classic methods. The VSMC method seems, at least in our knowledge, the first attempt to measure the proliferation in a very short period of time

5. Discussion: The discussion is not relevant i.e the comparison with resistant stains (this data was not shown by other methods in the methods and results)

Reply:

Resistant strains were originally identified by routine test. The discussion was deeply revised to answer the reviewer’s observation.

6. PLOS authors have the option to publish the peer review history of their article (what does this mean?). If published, this will include your full peer review and any attached files.

Reply: 

Not relevant for us.

Do you want your identity to be public for this peer review? For information about this choice, including consent withdrawal, please see our Privacy Policy.

Reviewer #1: No

Reviewer #2: No

---

## [Editor Report · Decision Letter 1]

28 Jan 2020

Evaluation of bacterial proliferation with a microfluidic-based device: Antibiochip.

PONE-D-19-26275R1

Dear Dr.Marco Pizzi

We are pleased to inform you that your manuscript has been judged scientifically suitable for publication and will be formally accepted for publication once it complies with all outstanding technical requirements.

With kind regards,

Shamala Devi Sekaran

Academic Editor

PLOS ONE
---

## [Editor Report · Acceptance letter]

5 Feb 2020

PONE-D-19-26275R1 

Evaluation of bacterial proliferation with a microfluidic-based device: Antibiochip. 

Dear Dr. Pizzi:

I am pleased to inform you that your manuscript has been deemed suitable for publication in PLOS ONE. Congratulations! Your manuscript is now with our production department. 

With kind regards,

on behalf of

Professor Shamala Devi Sekaran 

Academic Editor

PLOS ONE